# Qubit teleportation between non-neighbouring nodes in a quantum network

S. L. N. Hermans[1,3], M. Pompili[1,3], H. K. C. Beukers[1], S. Baier[1,2], J. Borregaard[1] & R. Hanson[1 ✉]

Future quantum internet applications will derive their power from the ability to share quantum information across the network[1,2]. Quantum teleportation allows for the reliable transfer of quantum information between distant nodes, even in the presence of highly lossy network connections[3]. Although many experimental demonstrations have been performed on different quantum network platforms[4–10], moving beyond directly connected nodes has, so far, been hindered by the demanding requirements on the pre-shared remote entanglement, joint qubit readout and coherence times. Here we realize quantum teleportation between remote, non-neighbouring nodes in a quantum network. The network uses three optically connected nodes based on solid-state spin qubits. The teleporter is prepared by establishing remote entanglement on the two links, followed by entanglement swapping on the middle node and storage in a memory qubit. We demonstrate that, once successful preparation of the teleporter is heralded, arbitrary qubit states can be teleported with fidelity above the classical bound, even with unit efficiency. These results are enabled by key innovations in the qubit readout procedure, active memory qubit protection during entanglement generation and tailored heralding that reduces remote entanglement infidelities. Our work demonstrates a prime building block for future quantum networks and opens the door to exploring teleportation-based multi-node protocols and applications[2,11–13].

Quantum teleportation is the central routine for reliably sending qubits across lossy network links[3], as well as a key primitive of quantum network protocols and applications[2,11,12]. Using a teleporter in the form of a pre-shared entangled state, the quantum information is transferred by performing a joint Bell-state measurement (BSM) on the sender's part of the entangled state and the qubit state to be teleported. The state is recovered on the receiving node by a gate operation conditioned on the BSM outcome[3]. Because the quantum information is not transmitted by a physical carrier, the protocol is insensitive to loss in the connecting photonic channels and on intermediate nodes. A deterministic BSM combined with real-time feed-forward enables unconditional teleportation, in which state transfer is achieved each time a qubit state is inserted into the teleporter.

Pioneering explorations of quantum teleportation protocols were performed using photonic states[4–6]. Following the development of quantum network nodes with stationary qubits, remote qubit teleportation was realized between trapped ions[7], trapped atoms[8,10], diamond nitrogen-vacancy (NV) centres[9] and memory nodes based on atomic ensembles[14].

Although future quantum network applications will widely use teleportation between non-connected nodes in the network, the demanding set of requirements on the pre-shared entanglement, the BSM and

the coherence times for enabling real-time feed-forward has, so far, prevented the realization of teleportation beyond directly connected stationary network nodes.

Here we overcome these challenges by a set of key innovations and achieve qubit teleportation between non-neighbouring network nodes (see Fig. 1a). Our quantum network consists of three nodes in a line configuration, Alice, Bob and Charlie. Each node contains a NV centre in diamond. Using the NV electronic spin as the communication qubit, we are able to generate remote entanglement between each pair of neighbouring nodes. In addition, Bob and Charlie each use a nearby [13]C nuclear spin as a memory qubit. The steps of the teleportation protocol are shown in Fig. 1b. To prepare the teleporter, we use an entanglement swapping protocol mediated by Bob, similar to a quantum repeater protocol[15], to establish entanglement between Alice and Charlie. Once successful preparation of the teleporter is heralded, the input qubit state is prepared on Charlie and finally teleported to Alice.

## Entanglement fidelity of the network links

A key parameter for quantum teleportation is the fidelity of the pre-shared entangled state between Alice and Charlie. As we generate this state by entanglement swapping, its fidelity can be increased by

[1]QuTech and Kavli Institute of Nanoscience, Delft University of Technology, Delft, The Netherlands. [2]Present address: Institut für Experimentalphysik, Universität Innsbruck, Innsbruck, Austria. [3]These authors contributed equally: S. L. N. Hermans, M. Pompili. ✉e-mail: R.Hanson@tudelft.nl

**Fig. 1 | Teleporting a qubit between non-neighbouring nodes of a quantum network. a**, Three network nodes, Alice (A), Bob (B) and Charlie (C), are connected by means of optical fibre links (lines) in a line configuration. Each setup has a communication qubit (purple) that enables entanglement generation with its neighbouring node. Furthermore, Bob and Charlie contain a memory qubit (yellow). **b**, The steps of the teleportation protocol. (1) We prepare the teleporter by establishing entanglement between Alice and Charlie using an entanglement swapping protocol on Bob, followed by swapping the state at Charlie to the memory qubit. (2) The qubit state to be teleported is prepared on the communication qubit on Charlie. (3) A BSM is performed on Charlie's qubits and the outcome is communicated to Alice over a classical channel. Dependent on this outcome, Alice applies a quantum gate to obtain the teleported qubit state.

mitigating errors on the individual links. Our network generates entanglement between neighbouring nodes using a single-photon protocol[16,17] in an optical-phase-stabilized architecture[18]. The building block of this protocol is a qubit–photon entangled state created at each node. To generate this entangled state, we initialize the communication qubit in a superposition state $|\psi\rangle = \sqrt{\alpha}\,|0\rangle + \sqrt{1-\alpha}\,|1\rangle$ and apply a state-selective optical pulse that transfers the population from $|0\rangle$ to an optically excited state. Following spontaneous emission, the qubit state is entangled with the photon number (0 or 1 photon). We perform this protocol on both nodes and interfere the resonant photonic states on a beam splitter (Fig. 2a). Detection of a single photon in one of the output ports ideally heralds the generation of an entangled state $|\psi\rangle = (|01\rangle \pm |10\rangle)/\sqrt{2}$, in which the ± phase is set by the detector that clicked. Figure 2b shows the joint outcomes of qubit measurements in the computational basis after entanglement is heralded, showing the expected correlations.

The infidelity of the generated state has three main contributions: double $|0\rangle$ state occupancy, double optical excitation and finite distinguishability of the photons[18,19]. In the case of double $|0\rangle$ state occupancy (which occurs with probability $\alpha$), both communication qubits are in the $|0\rangle$ state and have emitted a photon. Detection of one of these photons leads to false heralding of an entangled state. The second effect, double excitation, is due to the finite length of the optical pulse compared with the optical lifetime of the emitter. There is a finite chance that the communication qubit emits a photon during this pulse, is subsequently re-excited and then emits another photon, resulting in the qubit state being entangled with two photons. Detection or loss of the first photon destroys the coherence of the qubit–photon entangled state and detection of the second photon can then falsely herald the generation of an entangled state.

Crucially, false heralding events caused by double $|0\rangle$ state occupancy and double excitation are both accompanied by an extra emitted photon. Therefore, detection of this extra photon allows for unambiguous identification of such events and thus for real-time rejection of false heralding signals. We implement this rejection scheme by monitoring the off-resonant phonon-side band (PSB) detection path on both setups during and after the optical excitation (see Fig. 2a).

To investigate the effect of this scheme, we generate entanglement on the individual links and extract the entanglement heralding events for which the PSB monitoring flagged the presence of an extra photon. For these events, we analyse the corresponding qubit measurements in the computational basis (Fig. 2c).

We identify two separate regimes: one during the optical pulse (purple) and one after the optical pulse (yellow). When a photon is detected on Alice's (Bob's) PSB detector during the optical pulse, we see that the outcome 01 (10) is most probable (purple data in Fig. 2c), showing that only one setup was in the $|0\rangle$ state and thus that both detected photons

originated from Alice (Bob). The detection of PSB photons during the optical pulse thus primarily flags double excitation errors. By contrast, when a photon is detected after the optical pulse in either Alice's or Bob's PSB detector, the outcome 00 is most probable (yellow data in Fig. 2c), indicating that both setups were in the $|0\rangle$ state and emitted a photon. PSB photon detection after the optical pulse thus flags the double $|0\rangle$ state occupancy error. We find similar results to Fig. 2c for the entangled states generated on the Bob–Charlie link; see Extended Data Fig. 2. The improvement in fidelity from rejecting these false heralding events in our experiment is set by the combined probability of occurrence (≈9%; see Supplementary Information) multiplied by the probability to flag them (given here by the total PSB photon detection efficiency of ≈10%).

The third main source of infidelity, the finite distinguishability, can arise from frequency detunings between the emitted photons[20]. Whereas most of these detunings are eliminated up front by the charge-resonance (CR) check before the start of the protocol (see Supplementary Information), the communication qubits may still be subject to a small amount of spectral diffusion. In our single-photon protocol, this leads to dephasing that is stronger for photons that are detected later relative to the optical pulse. By shortening our detection window, we can increase the fidelity of the entangled state at the expense of a lower entangling rate. For the experiments below (unless mentioned otherwise), we use a detection window length of 15 ns. Figure 2d summarizes the measured improvements on the individual links and the estimated effect on the Alice–Charlie entangled state fidelity. The increase of ≈3% is instrumental in pushing the teleportation fidelity above the classical bound.

## Memory qubit coherence

In the preparation of the teleporter, we reliably preserve the Alice–Bob entangled link on the memory qubit, by aborting the sequence and starting over when the Bob–Charlie entangled state is not heralded within a fixed number of attempts, the timeout.

The $^{13}$C memory qubits can be controlled with high fidelity by means of the communication qubit, although they can be efficiently decoupled when no interaction is desired. Recent work showed that, in a magnetic field of 189 mT, entanglement generation attempts with the communication qubit do not limit the memory dephasing time $T_2^*$ (ref. [18]), opening the door to substantially extending the memory preservation time with active coherence protection from the spin bath[21]. We realize this protection by integrating a decoupling π-pulse on the memory qubit into the experimental sequence that follows a heralding event, while ensuring that all phases that are picked up owing to the probabilistic nature of the remote entangling process are compensated in real time (Fig. 3a).

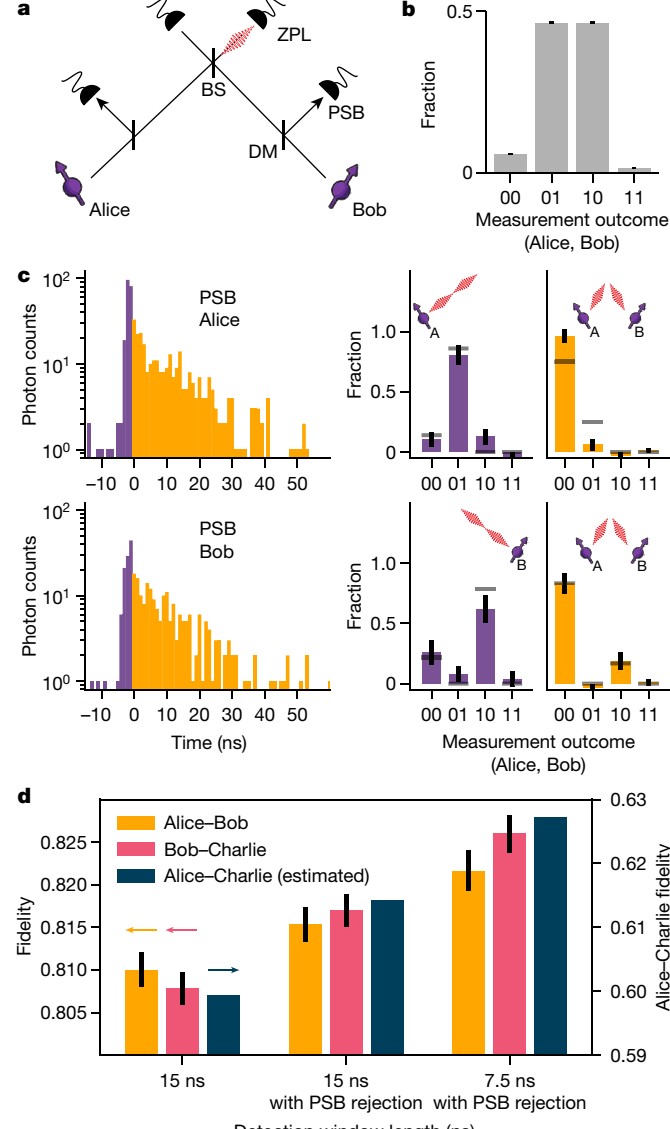

**a**

ZPL

BS

PSB

DM

Alice

Bob

**b**

**c**

PSB
Alice

PSB
Bob

**d**

Fig. 2 | **High-fidelity entangled network links. a**, Simplified schematic of the optical link used for generating entanglement between neighbouring nodes. Photons emitted by the communication qubits are filtered by a dichroic mirror (DM) to separate the resonant (zero-phonon line, ZPL) photons (3% of emission) from the off-resonant (phonon-side band, PSB) photons (97% of emission). The resonant photons are sent to the beam splitter (BS); detection of a single photon at one of the ZPL detectors heralds successful generation of an entangled state between the two nodes. **b**, Measured correlations of the communication qubits in the computational basis, conditioned on a heralding event on the ZPL detectors. **c**, Left, histograms of the PSB photon detection times on Alice (top) or Bob (bottom), conditioned on a simultaneous ZPL detection in the same entanglement generation attempt. Grey lines show expected correlations on the basis of a quantum-optical model (see Supplementary Information). The correlations measured in the other measurement bases can be found in Extended Data Fig. 1. **d**, Measured fidelity of the network links, without PSB rejection (left), with PSB rejection (middle) and with PSB rejection plus shortened detection window (right). The dark blue bars indicate the corresponding expected fidelity on Alice–Charlie after entanglement swapping for each case (see Methods). All error bars represent one standard deviation.

In Fig. 3b, we check the performance of this sequence by storing a superposition state on the memory qubit and measuring the Bloch vector length. We observe that, without the decoupling pulse, the decay of the Bloch vector length is not altered by the entanglement

attempts, in line with previous findings[18]. By contrast, when we apply the decoupling pulse, the decay is slowed down by more than a factor of 6, yielding a $N_{1/e}$ decay constant of ≈5,300 entanglement attempts, the highest number reported so far for diamond devices. The difference in the shape of the decay indicates that intrinsic decoherence is no longer the only limiting factor. The improved memory coherence enables us to use a timeout of 1,000 entangling attempts, more than double that of ref. [18], which doubles the entanglement swapping rate.

## Memory qubit readout

High-fidelity memory qubit readout is required both in the preparation of the teleporter (at Bob) and during the teleportation protocol itself (at Charlie). The memory qubit is read out by mapping its state onto the communication qubit using quantum logic followed by single-shot readout of the communication qubit using state-dependent optical excitation and detection[22]. Owing to limited photon collection efficiency (≈10%) and finite cyclicity of the optical transition (≈99%), the communication qubit readout fidelity is different for |0⟩ and |1⟩ and the probability that the correct state was assigned is much larger if one or more photons were detected (assigned outcome 0) than if no photons were detected (assigned outcome 1)[23]. In previous work, we circumvented this issue by conditioning on obtaining the outcome 0 (ref. [18]). However, this approach scales unfavourably, as it forces the protocol to prematurely abort with probability >50% at each memory qubit readout.

We resolve this challenge by introducing a basis-alternating repetitive readout for the memory qubit (see Fig. 3c). The key point of this readout strategy is, in contrast to earlier work[24], to alternately map the computational basis states of the memory qubit to the communication qubit state |0⟩ Figure 3d shows the readout fidelities of the $n$th readout repetition for the two initial states for the memory qubit on Bob (for Charlie, see Extended Data Fig. 3). We clearly observe the expected alternating pattern owing to the asymmetry of the communication qubit readout fidelities. Notably, the readout fidelity decays only by ≈1% per readout, showing that the readout is mostly non-demolition and several readouts are possible without losing the state.

Next, we assign the state using the first readout and continue the sequence only when the consecutive readouts are consistent with the first readout. The subsequent readouts therefore add confidence to the assignment in the case of consistent outcomes, whereas cases of inconsistent outcomes (which have a higher chance of indicating an incorrect assignment) are filtered out. In Fig. 3e, we plot the readout fidelity resulting from this strategy for up to five readouts, with the corresponding rejected fraction due to inconsistent outcomes plotted in Fig. 3f. We observe that using two readouts already eliminates most of the asymmetry, reducing the average infidelity from ≈6% to below 1%. At this point, the remaining observed infidelity mainly results from cases in which the memory qubit was flipped during the first readout block because of imperfect memory qubit gates. For the experiments reported below (unless mentioned otherwise), we use two readout repetitions to benefit from a high average readout fidelity (Bob: 99.2(4)%, Charlie: 98.1(4)%) and a high probability to continue the sequence (Bob and Charlie: ≈88%).

## Teleporting qubit states from Charlie to Alice

With all innovations described above implemented, we perform the protocol as shown in Fig. 4a. First, we generate entanglement between Alice and Bob and store Bob's part of the entangled state on the memory qubit using a compiled SWAP operation. Second, we generate entanglement between Bob and Charlie, while preserving the first entangled state on the memory qubit with the pulse sequence as described in Fig. 3a. Next, we perform a BSM on Bob followed by a CR check. We continue the sequence if the communication qubit readout yields outcome 0, the memory qubit readout gives a consistent outcome pattern and the CR check is passed. At Charlie, we perform a quantum gate that depends

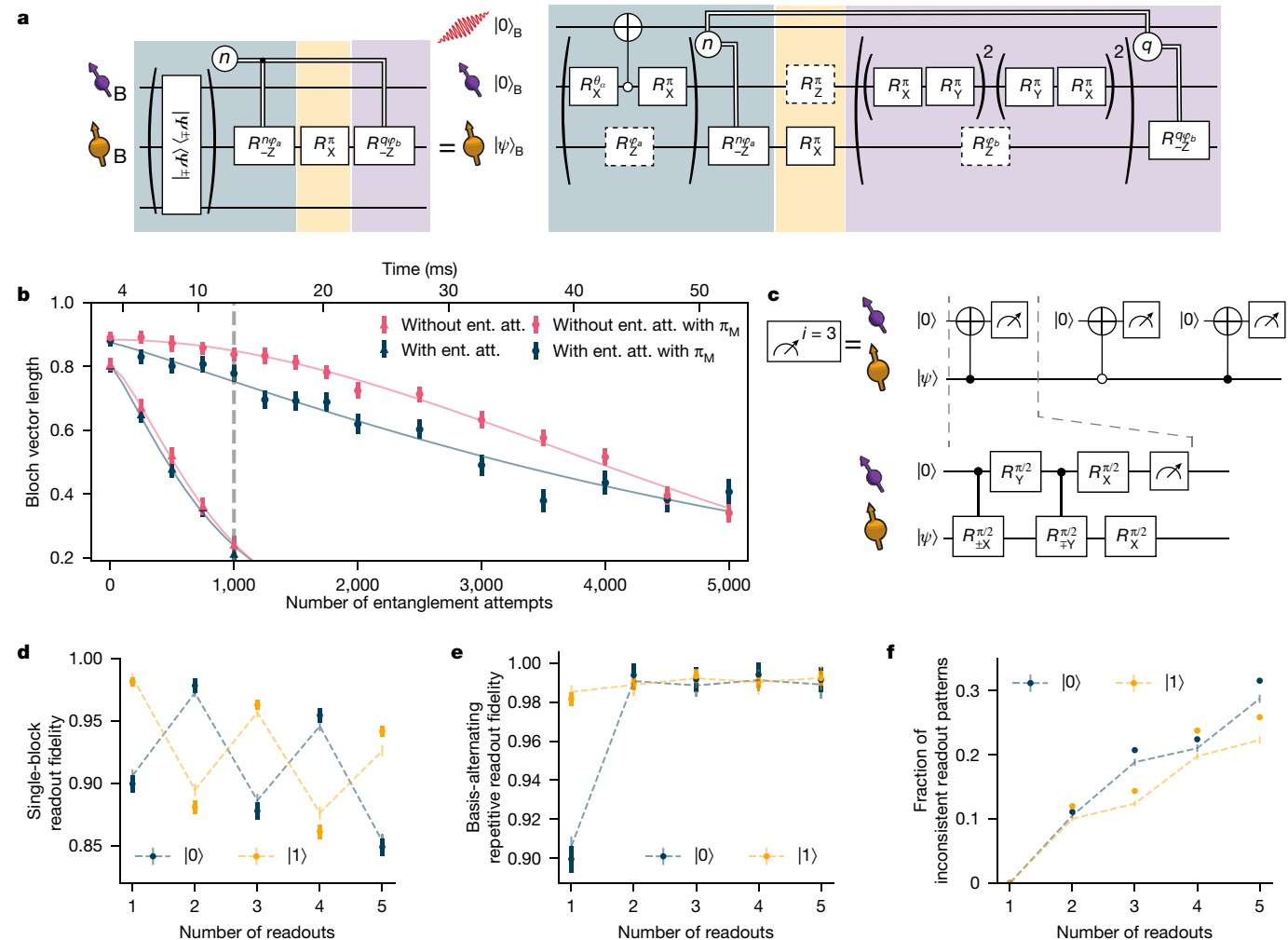

**Fig. 3 | Memory qubit coherence and readout. a**, Gate sequence on Bob for entanglement generation with the communication qubit while preserving states stored on the memory qubit. Entanglement generation attempts are repeated until success or a predetermined timeout. On success in the $n$th attempt, a phase feed-forward is applied to maintain the correct reference frame of the memory qubit[18], followed by a decoupling pulse on the memory qubit. The decoupling $\pi_M$ pulse causes a Z rotation on the communication qubit. Afterwards, we rephase the memory qubit for the same amount of time as it took to herald entanglement (by applying $q$ blocks of XY8 decoupling sequences on the communication qubit, in which $q$ depends on the number of entanglement attempts needed $n$) and we end with another phase feed-forward on the memory qubit, to compensate for any phase picked up during this decoupling. **b**, Bloch vector length of a superposition state stored on the memory qubit for different number of entanglement attempts or a time-equivalent wait element. In the case of no decoupling (no $\pi_M$) on the memory qubit, the gates in the yellow shaded box in **a** are left out. The grey dashed line indicates the chosen timeout of 1,000 entanglement attempts. **c**, Gate sequence for the basis-alternating repetitive readout of the memory qubit. **d**, Readout fidelity for each readout repetition, for states $|0\rangle$ and $|1\rangle$. **e**, Readout fidelity of the basis-alternating repetitive readout scheme for different number of readout repetitions. **f**, Fraction of inconsistent readout patterns for different number of readout repetitions. In **d**–**f**, the dashed lines show a numerical model using measured parameters. All error bars represent one standard deviation.

on the outcome of the BSM and on which detectors clicked during the two-node entanglement generation. Next, we swap the entangled state to the memory qubit. At this point, the teleporter is ready and Alice and Charlie share an entangled state with an estimated fidelity of 0.61.

Subsequently, we generate the qubit state to be teleported, $|\psi\rangle$, on Charlie's communication qubit and run the teleportation protocol. First, a BSM is performed on the communication and memory qubits at Charlie. With the exception of unconditional teleportation (discussed below), we only continue the sequence when we obtain a 0 outcome on the communication qubit, when we have a consistent readout pattern on the memory qubit and when Charlie passes the CR check. The outcomes of the BSM are sent to Alice and, by applying the corresponding gate operation, we obtain $|\psi\rangle$ on Alice's side.

We teleport the six cardinal states ($\pm X$, $\pm Y$, $\pm Z$), which form an unbiased set[25], and measure the fidelity of the teleported states to the ideally prepared state (Fig. 4b). We find an average teleported state fidelity of $F = 0.702(11)$ at an experimental rate of $1/(117\ \mathrm{s})$. This value exceeds the classical bound of 2/3 by more than three standard deviations, thereby proving the quantum nature of the protocol. We note that this value provides a lower bound to the true teleportation fidelity, as the measured fidelity is decreased by errors in the preparation of the qubit states at Charlie (estimated to be 0.5%; see Supplementary Information).

The differences in fidelity between the teleported states arise from an interplay of errors in different parts of the protocol that either affect all three axes (depolarizing errors) or affect only two axes (dephasing errors). These differences are qualitatively reproduced by our model (grey bars in Fig. 4b). In Fig. 4c, we plot the teleportation fidelity for each possible outcome of the BSM. Owing to the basis-alternating repetitive readout, the dependence on the second bit (from the memory qubit readout) is small, whereas for the first bit (communication qubit readout), the best teleported state fidelity is achieved for outcome 0, due to the asymmetric readout fidelities. We also analyse the case in which no

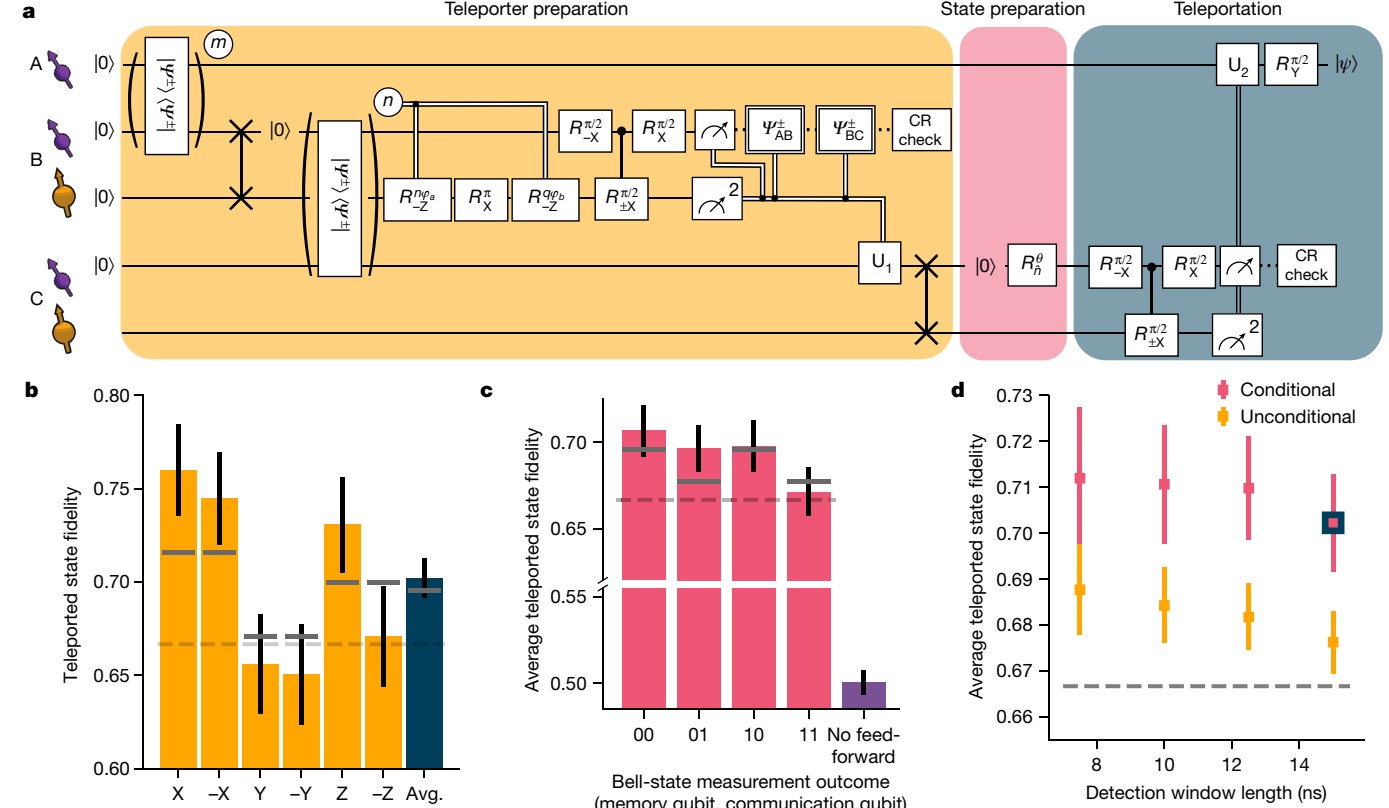

**Fig. 4 | Qubit teleportation between non-neighbouring network nodes. a**, Circuit diagram of the teleportation protocol using notation defined in Fig. 3. $m$ ($n$) is the number of attempts needed to herald entanglement for the AB (BC) entangled link. See the Supplementary Information for the full circuit diagram. **b**, Teleported state fidelities for the six cardinal states and their average (Avg.). The grey lines show the expected fidelities from simulations. The dashed lines in **b**–**d** represent the classical bound of 2/3. **c**, Average teleported state fidelity for the different outcomes of the BSM on Charlie. The right-most bar shows the resulting fidelity when no feed-forward operation on Alice would be applied. The numerical values of the bar plots shown in **b** and **c** can be found in Extended Data Tables 1 and 2. **d**, Average state fidelity for a conditional and an unconditional teleportation, for different detection window lengths of the two-node entanglement generation processes. The blue-bordered data point is the same point as shown in **b**. All error bars represent one standard deviation.

feed-forward is applied at Alice (see Methods); as expected, the average state fidelity reduces to a value consistent with a fully mixed state (fidelity $F = 0.501(7)$), emphasizing the critical role of the feed-forward in the teleportation protocol.

Finally, we demonstrate that the network can achieve unconditional teleportation between Alice and Charlie by using the BSM in a deterministic fashion. To this end, we revise the protocol at Charlie to accept both communication qubit outcomes, use all memory qubit readout patterns, including the inconsistent ones, and disregard the outcome of the CR check after the BSM. Using this fully deterministic BSM lowers the average teleportation fidelity by a few percent (Fig. 4d). At the same time, shortening the detection windows of the two-node entanglement generation is expected to yield an improvement in the fidelity, as discussed above. We find that, indeed, the average unconditional teleportation fidelity increases with shorter window lengths, reaching $F = 0.688(10)$ for a length of 7.5 ns and a rate of 1/(100 s); see Extended Data Fig. 4. The current quantum network is thus able to perform teleportation beyond the classical bound, even under the strict condition that every state inserted into the teleporter be transferred.

## Outlook

In this work, we have realized unconditional qubit teleportation between non-neighbouring nodes in a quantum network. The innovations introduced here on memory qubit readout and protection during entanglement generation, as well as the real-time rejection of false heralding signals, will be instrumental in exploring more complex

protocols[2,11–13,26]. Also, these methods can be readily transferred to other platforms, such as the group IV colour centres in diamond, the vacancy-related qubits in SiC and single rare-earth ions in solids[27–33].

The development of an improved optical interface for the communication qubit[34] will increase both the teleportation protocol rate and fidelity. Because of the improved memory qubit performance reported here, the network already operates close to the threshold at which nodes can reliably deliver a remote entangled state while preserving previously stored quantum states in their memory qubits. With further improvements, for instance, by integrating multi-pulse memory decoupling sequences[21] into the entanglement generation, demonstration of deterministic qubit teleportation (with no pre-shared entangled state) may come within reach, which opens the door to exploring applications that call the teleportation routine several times. In addition, future work will focus on further improving the phase stabilization and extending the current schemes for use in deployed fibre[35].

Finally, by implementing a recently proposed link layer protocol[36], qubit teleportation and applications making use of the teleportation primitive may be executed and tested on the network through platform-independent control software, an important prerequisite for a large-scale future network.

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

## Methods

### Experimental setup

The basics of the experimental setup are described in ref. [18]. In the current experiment, Charlie has access to a carbon-13 nuclear spin that acts as a memory qubit. The parameters used for the memory qubits of Bob and Charlie can be found in Extended Data Table 3. Furthermore, we have set up a classical communication channel between Charlie and Alice, such that Charlie can directly send the results of the BSM to Alice.

### Temporal selection of heralding photons

To eliminate any reflected excitation light in the heralding detectors, we make use of a cross-polarization scheme and perform temporal selection of the detected photons as described in ref. [37]. We start the detection windows 4 ns (5 ns) after the highest intensity point of the excitation pulse, for the AB (BC) entangled link, to ensure sufficient suppression of excitation laser light in the detection window.

### Memory qubit coherence Bob

We use the sequence described in Fig. 3a to preserve the state of the memory qubit during entanglement attempts. To characterize the decoupling sequence, we compare it to the sequence in which we do not apply the decoupling pulse on the memory qubit and/or the sequence in which we idle instead of performing entanglement attempts. We characterize the coherence of the memory qubit by storing the six cardinal states. We average the results for the eigenstates ($|0\rangle$, $|1\rangle$) and superposition states ($|\pm X\rangle$ and $|\pm Y\rangle$). In Extended Data Fig. 5a, we plot the Bloch vector length $b = \sqrt{b_x^2 + b_y^2 + b_z^2}$, with $b_i$ the Bloch vector component in direction $i$.

Over the measured range, the eigenstates show little decay. The decay of the superposition states is fitted with the function $f(x) = Ae^{-(x/N_{1/e})^n}$. The fitted parameters can be found in Extended Data Fig. 5b.

The use of the decoupling pulse $\pi_M$ on the memory qubit increases the $N_{1/e}$ by more than a factor of 6. Moreover, the initial Bloch vector length $A$ is higher with the $\pi_M$ pulse. This is mainly explained by the second round of phase stabilization[18] in between swapping the state onto the memory qubit and starting the entanglement generation process. The phase stabilization takes $\approx 350$ μs and, during this time, the memory qubit is subject to intrinsic $T_2^*$ dephasing, which can be efficiently decoupled using the $\pi_M$ pulse.

### Communication qubit coherence

In various parts of the protocol, we decouple the communication qubits from the spin bath environment to extend their coherence time. On Alice, we start the decoupling when the first entangled link is established and stop when the results of the BSM to teleport the state are sent by Charlie. On Bob, we decouple the communication qubit when the memory qubit is being rephased. On Charlie, the communication qubit is decoupled from the point that entanglement with Bob is heralded up to the point at which Bob has finished the BSM, performed the CR check and has communicated the results. All these decoupling times are dependent on how many entanglement attempts are needed to generate the entangled link between Bob and Charlie.

We characterize the average state fidelities for different decoupling times; see Extended Data Fig. 6a. We investigate eigenstates and superposition states separately. We fit the fidelity with the function $f(t) = Ae^{-(t/\tau_{coh})^n} + 0.5$. The fitted parameters are summarized in Extended Data Fig. 6b. For each setup, the minimum and maximum decoupling times used are indicated by the shaded regions in Extended Data Fig. 6a. The left-most border is the decoupling time when the first entanglement attempt on Bob and Charlie would be successful and the right-most border is when the last attempt before the timeout of 1,000 attempts would herald the entangled state.

### Model of the teleported state

A detailed model of the teleported state can be found at https://doi.org/10.4121/16645969. The model comprises elements from ref. [18] and is further extended for the teleportation protocol. We take the following noise sources into account:

- Imperfect Bell states between Alice and Bob, and between Bob and Charlie.
- Dephasing of the memory qubit of Bob during entanglement generation between Bob and Charlie.
- Depolarizing noise on the memory qubits of Bob and Charlie, owing to imperfect initialization and swap gates.
- Readout errors on the communication qubits of Bob and Charlie and readout errors on the memory qubits of Bob and Charlie when using the basis-alternating readout scheme, which result in incorrect feed-forward gate operations after the BSMs.
- Depolarizing noise on Alice during the decoupling sequence.
- Ionization probability on Alice.

An overview of the input parameters and the effect of the different error sources are given in Extended Data Table 4.

### Calculation of teleported state fidelity without feed-forward operation

In Fig. 4c, we show the fidelity of the teleported state in case no feed-forward operations would have been applied on Alice. To extract this data, we follow the same method as in ref. [9]. We perform classical bit flips on the measurement outcomes to counteract the effect of the feed-forward gate operations (as if the gate was not applied) for each BSM outcome. We do this for all six cardinal states and compute the average fidelity. We assume the errors of the gate in the feed-forward operations to be small.

## Data availability

The datasets that support this manuscript and the software to analyse them are available at https://doi.org/10.4121/16645969.

37. Hensen, B. et al. Loophole-free Bell inequality violation using electron spins separated by 1.3 kilometres. *Nature* **526**, 682–686 (2015).

**Acknowledgements** We thank S. Wehner, T. Taminiau, C. Bradley and H. de Riedmatten for discussions. We acknowledge financial support from the EU Flagship on Quantum Technologies through the project Quantum Internet Alliance (EU Horizon 2020, grant agreement no. 820445); from the European Research Council (ERC) through an ERC Consolidator Grant (grant agreement no. 772627 to R.H.); from the Netherlands Organisation for Scientific Research (NWO) through a VICI grant (project no. 680-47-624) and the Zwaartekracht program Quantum Software Consortium (project no. 024.003.037/3368). S.B. acknowledges support from an Erwin-Schrödinger fellowship (QuantNet, no. J 4229-N27) of the Austrian National Science Foundation (FWF).

**Author contributions** S.L.N.H., M.P. and R.H. devised the experiment. S.L.N.H., M.P. and H.K.C.B. carried out the experiments and collected the data. S.L.N.H., M.P., H.K.C.B. and S.B. prepared the experimental apparatus. J.B. developed the quantum-optical model. S.L.N.H. and R.H. wrote the main manuscript, with input from all authors. S.L.N.H., M.P. and J.B. wrote the supplementary materials, with input from all authors. S.L.N.H. and M.P. analysed the data and discussed with all authors. R.H. supervised the research.

**Competing interests** The authors declare no competing interests.

**Additional information**
**Correspondence and requests for materials** should be addressed to R. Hanson.

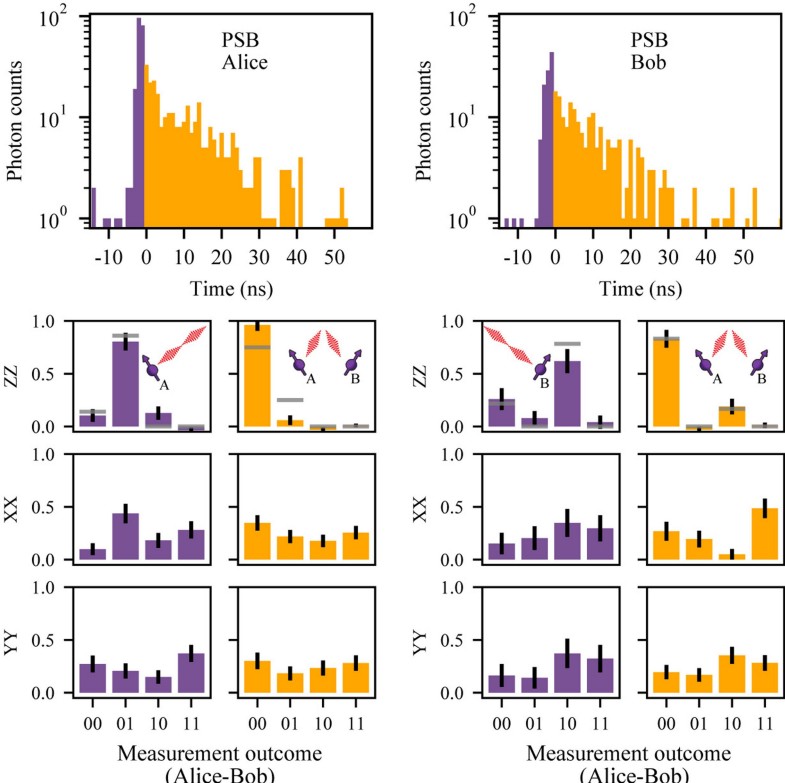

**Extended Data Fig. 1 | PSB-flagged correlations Alice–Bob.** Top, histograms of the detected PSB photons conditioned on a simultaneous ZPL detection in the entanglement generation attempt, for Alice (left) and Bob (right). Bottom, corresponding measured correlations in all bases. The grey bars in the Z basis represent the simulated values. For the X and Y bases, one would expect a probability of 0.25 for all outcomes. All error bars represent one standard deviation.

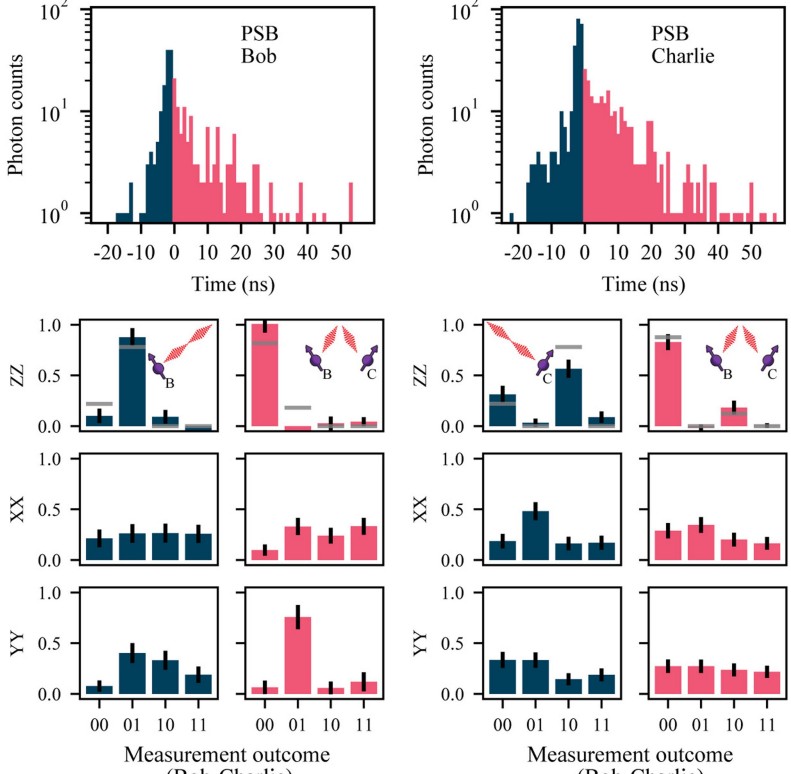

**Extended Data Fig. 2 | PSB-flagged correlations Bob–Charlie.** Top, histograms of the detected PSB photons conditioned on a simultaneous ZPL detection in the entanglement generation attempt, for Bob (left) and Charlie (right). Bottom, corresponding measured correlations in all bases. The grey bars in the Z basis represent the simulated values. For the X and Y bases, one would expect a probability of 0.25 for all outcomes. All error bars represent one standard deviation.

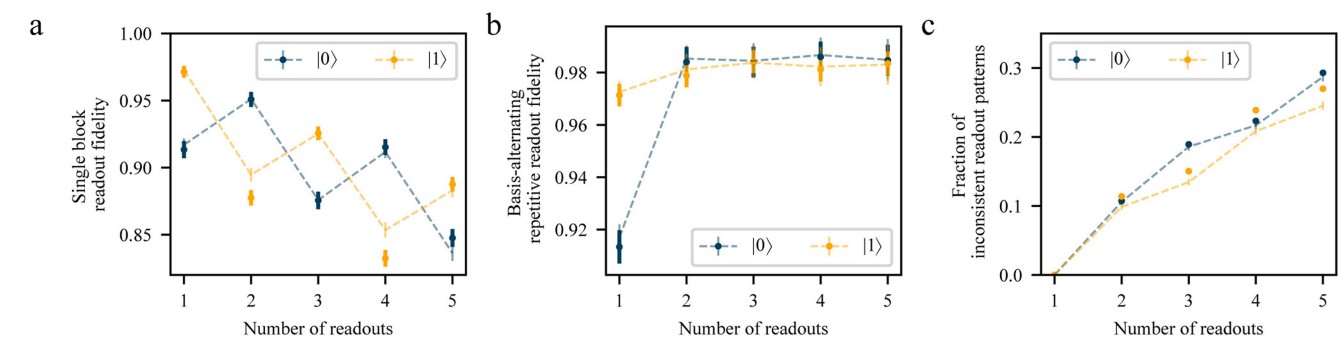

**Extended Data Fig. 3 | Basis-alternating repetitive readout.**
Basis-alternating repetitive (BAR) readout results for Charlie's memory qubit.
**a**, Readout fidelity for each readout repetition, for states |0⟩ and |1⟩. **b**, Readout
fidelity of the BAR readout scheme for different number of readout repetitions.
**c**, Fraction of inconsistent readout patterns for different number of readout
repetitions. The dashed lines represent a numerical model using measured
parameters, which can be found at https://doi.org/10.4121/16645969. All error
bars represent one standard deviation.

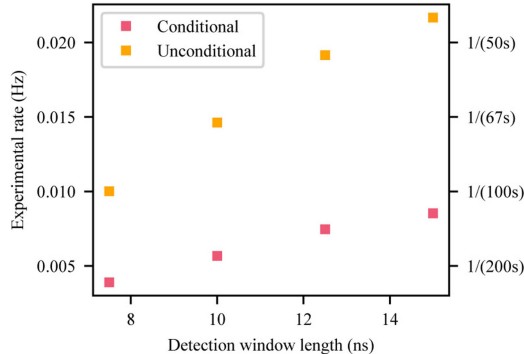

**Extended Data Fig. 4 | Experimental rates.** Experimental rates of the conditional and unconditional teleportation protocol for different detection window lengths in the two-node entanglement generation.

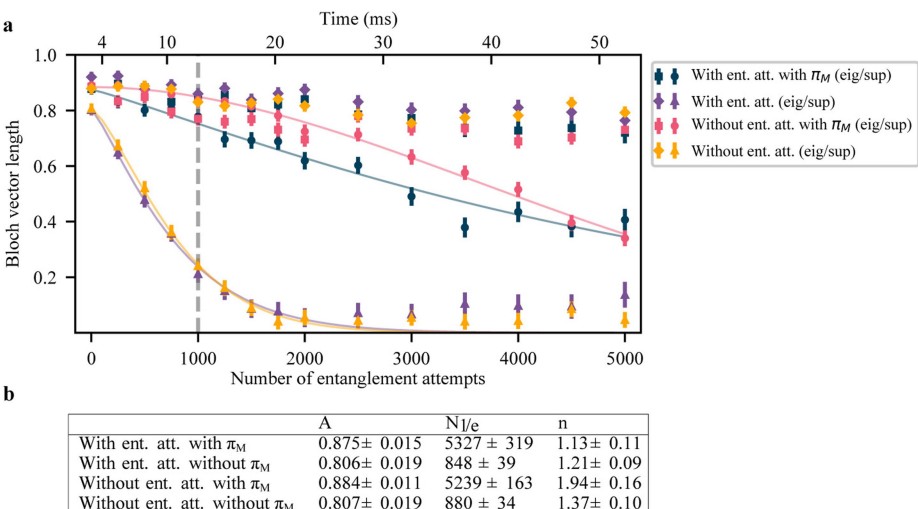

**a**

**b**

| | A | N$_{1/e}$ | n |
|---|---|---|---|
| With ent. att. with $\pi_M$ | 0.875± 0.015 | 5327 ± 319 | 1.13± 0.11 |
| With ent. att. without $\pi_M$ | 0.806± 0.019 | 848 ± 39 | 1.21± 0.09 |
| Without ent. att. with $\pi_M$ | 0.884± 0.011 | 5239 ± 163 | 1.94± 0.16 |
| Without ent. att. without $\pi_M$ | 0.807± 0.019 | 880 ± 34 | 1.37± 0.10 |

**Extended Data Fig. 5 | Memory qubit coherence. a**, Coherence of Bob's memory qubit for superposition states (triangles and circles) and eigenstates (squares and diamonds). We perform the sequence as described in the main text with and without the decoupling pulse $\pi_M$ on the memory qubit, the dark blue and purple points, respectively. Furthermore, we perform the sequence with a wait time instead of entanglement attempts with (pink points) and without (yellow points) the decoupling pulse. The grey dashed line indicates the timeout of the entanglement generation process used in the teleportation protocol. **b**, Fitted parameters for the memory coherence decay of the superposition states. All error bars represent one standard deviation.

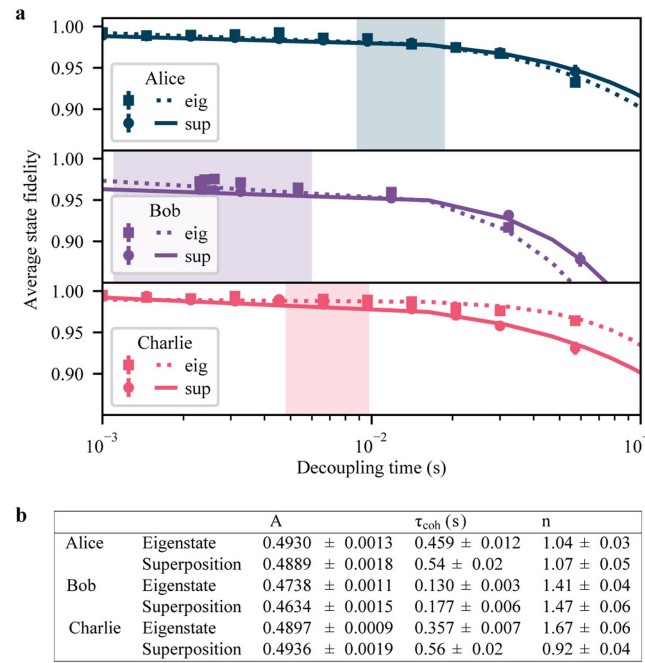

**b**

| | | A | $\tau_{coh}$ (s) | n |
|---|---|---|---|---|
| Alice | Eigenstate | 0.4930 ± 0.0013 | 0.459 ± 0.012 | 1.04 ± 0.03 |
| | Superposition | 0.4889 ± 0.0018 | 0.54 ± 0.02 | 1.07 ± 0.05 |
| Bob | Eigenstate | 0.4738 ± 0.0011 | 0.130 ± 0.003 | 1.41 ± 0.04 |
| | Superposition | 0.4634 ± 0.0015 | 0.177 ± 0.006 | 1.47 ± 0.06 |
| Charlie | Eigenstate | 0.4897 ± 0.0009 | 0.357 ± 0.007 | 1.67 ± 0.06 |
| | Superposition | 0.4936 ± 0.0019 | 0.56 ± 0.02 | 0.92 ± 0.04 |

**Extended Data Fig. 6 | Communication qubit coherence. a**, Decoupling of the communication qubits. The average state fidelity is plotted for different decoupling times for each setup. The shaded area represents the decoupling times used in the teleportation protocol. **b**, Fitted parameters for average state fidelity during communication qubit decoupling. All error bars represent one standard deviation.

**Extended Data Table 1 | Teleported state fidelities**

|  | Teleported state fidelity |
|---|---|
| X | $0.760 \pm 0.024$ |
| -X | $0.745 \pm 0.025$ |
| Y | $0.656 \pm 0.027$ |
| -Y | $0.651 \pm 0.027$ |
| Z | $0.731 \pm 0.026$ |
| -Z | $0.671 \pm 0.027$ |
| Average | $0.702 \pm 0.011$ |

Numerical values of the data shown in Fig. 4b. All error bars represent one standard deviation.

**Extended Data Table 2 | Average teleported state fidelities per BSM outcome**

| Bell-state measurement outcome (memory qubit, communication qubit) | Average teleported state fidelity |
|---|---|
| 00 | $0.707 \pm 0.015$ |
| 01 | $0.696 \pm 0.014$ |
| 10 | $0.698 \pm 0.015$ |
| 11 | $0.671 \pm 0.014$ |
| No feed forward | $0.501 \pm 0.007$ |

Numerical values of the data shown in Fig. 4c. All error bars represent one standard deviation.

**Extended Data Table 3 | Memory qubit characteristics**

|  | Bob | Charlie |
|---|---|---|
| $B_z$ | 1890 Gauss | 165 Gauss |
| $\omega_{m_s=0}$ | $2\pi \times 2025$ kHz | $2\pi \times 177$ kHz |
| $\omega_{m_s=-1}$ | $2\pi \times 2056$ kHz | $2\pi \times 240$ kHz |
| $A_\parallel$ | $2\pi \times 30$ kHz | $2\pi \times 63$ kHz |
| $\tau_{con}$ | 2.818 $\mu s$ | 6.003 $\mu s$ |
| $N_{\pi/2,\mathrm{con}}$ | 54 | 56 |
| $\tau_{unc}$ | 4.165 $\mu s$ | 11.996 $\mu s$ |
| $N_{\pi/2,\mathrm{unc}}$ | 144 | 30 |

In each setup, we use a magnetic field with strength $B_z$ aligned to the NV axis. The nuclear spin precession frequencies ($\omega_{m_s=0}$ and $\omega_{m_s=-1}$) depend on the electron spin state. From the frequency difference, the parallel component $A_\parallel$ of the hyperfine interaction can be estimated. Conditional (unconditional) pulses are applied by $N_{\pi/2con}$ ($N_{\pi/2unc}$) pulses on the electron spin with an inter-pulse delay of $\tau_{con}$ ($\tau_{unc}$).

**Extended Data Table 4 | Two-node and teleportation simulation parameters**

| | Parameter AB | Parameter BC | Infidelity $|\Psi_{AB}\rangle$ | Infidelity $|\Psi_{BC}\rangle$ |
|---|---|---|---|---|
| Detection window length | 15 ns | 15 ns | | |
| Detection probability setup 1 | $3.4\times10^{-4}$ | $4.3\times10^{-4}$ | | |
| Detection probability setup 2 | $5.1\times10^{-4}$ | $2.4\times10^{-4}$ | | |
| Average detection probability PSB | 0.10 | 0.12 | | |
| $|0\rangle$ state populations $(\alpha_1, \alpha_2)$ | (0.07, 0.05) | (0.05, 0.1) | $5.5 \times10^{-2}$ | $6.7 \times10^{-2}$ |
| Dark count rate | 10 Hz | 10 Hz | $5.1 \times10^{-3}$ | $5.3 \times10^{-3}$ |
| Visibility | 0.90 | 0.90 | $2.4 \times10^{-2}$ | $2.4 \times10^{-2}$ |
| Average double excitation probability | 0.06 | 0.08 | $5.5 \times10^{-2}$ | $7.1 \times10^{-2}$ |
| Optical phase uncertainty | $21^o$ | $12^o$ | $3.1 \times10^{-2}$ | $1.0 \times10^{-2}$ |
| All error sources combined | | | 0.16 | 0.17 |

| | Parameter Teleportation | Infidelity $|\psi\rangle$ |
|---|---|---|
| Ionization probability Alice | 0.7% | $0.6 \times10^{-2}$ |
| Depolarizing noise Alice | 0.04 | $1.7 \times10^{-2}$ |
| Depolarizing noise memory qubit Bob | 0.12 | $5.0 \times10^{-2}$ |
| Dephasing noise memory qubit Bob $(N_{1/e}, n)$ | (5300, 1.1) | $2.1 \times10^{-2}$ |
| Depolarizing noise memory qubit Charlie | 0.14 | $5.9 \times10^{-2}$ |
| Readout fidelities memory qubit Bob $(|0\rangle, |1\rangle)$ | (0.99, 0.99) | $0.6 \times10^{-2}$ |
| Readout fidelities communication qubit Bob $(|0\rangle, |1\rangle)$ | (0.93, 0.995) | $0.3 \times10^{-2}$ |
| Readout fidelities memory qubit Charlie $(|0\rangle, |1\rangle)$ | (0.98, 0.98) | $1.1 \times10^{-2}$ |
| Readout fidelities communication qubit Charlie $(|0\rangle, |1\rangle)$ | (0.92, 0.99) | $0.6 \times10^{-2}$ |
| Two-node entangled states combined | | 0.192 |
| All error sources combined | | 0.305 |

Overview of parameters used in the simulations for the two-node entangled states and the teleported state. The two-node entangled states have an inherent error as a result of the single-photon entanglement protocol. For the other error sources, we compute the estimated infidelity as if it was the only error source present apart from the protocol error. We simulate the average teleported state fidelity in the case of a conditional BSM on Charlie. Again, for each error source, we compute the estimated infidelity as if it was the only error source present apart from the single-photon protocol errors of the two-node entangled states. This allows easy comparison between the different error sources.