## [Peer Review File · Nature]

Manuscript Title: Qubit teleportation between non-neighboring nodes in a quantum network

Reviewer Comments & Author Rebuttals

Reviewer Reports on the Initial Version:

Referees' comments:

Referee #1 (Remarks to the Author):

The report describes a protocol for entangling remote nodes of a quantum network by way of an entanglement swapping procedure at a third (center node). The distributed entanglement is then used for a teleportation operation between the remote nodes. The enabling features of this protocol are the novel quantum engineering schemes used to preserve the fidelity of the distributed entanglement at a critical level for teleportation to be performed above the classical bound. These novel schemes include techniques for removing contributions to the infidelity (false heralding events) of the generated state, preserving adjacent node entanglement long enough for both local links to be fully prepared using efficient and controllable quantum memories at one of the local links. Another key enabling feature is the non-demolition read-out of the quantum memory, thereby preserving the state stored in quantum memory over several readout attempts, thus allowing more attempts to complete a successful entangled link. Subsequently, the authors provide a higher fidelity teleportation ($F = 0.702$ - lower bounded by characterized errors) for a non-deterministic implementation of the protocol and at a slightly lower fidelity ($F=0.688$) and lower rate for a deterministic implementation whereby each communication photon inserted into the network is teleported.

The results are well presented, and the key features/schemes used are well characterized. This helps greatly to follow the progression of the protocol and to see how incremental improvements in performances of various steps in the protocol due to these techniques enable the distributed entanglement to be suitable for teleportation. Importantly, the impact of these innovations (eg applying a decoupling pi-pulse to actively protect the coherence of the memory; feed-forward in the teleportation protocol) are validated by following the particular steps with and without them.

I just have a few questions or clarifications. The technique at the end of the day is similar to a quantum repeater protocol - yet this term is not used. Is there a reason for this or do the authors disagree that the process represents a quantum repeater protocol? Why not refer to this as a quantum repeater protocol since it creates multiple elementary links (Alice to Bob and Bob to Charlie) and then employs entanglement swapping by way of a Bell State Measurement (BSM) at Bob to extend the entanglement to the further away nodes at Alice and Charlie?

Can you detail what percentage of false heralding events are rejected in the scheme using phonon-side band detection? Are there situations where the second photon (the signature of the falsity) can go undetected (other than the obvious failure of the detection event itself). In other words, can the

second photon 'hide' (perhaps by exiting the BS on the same path as the first photon) so as to be undetected and thus this scheme does not flag them? If so – what impact can this have on the overall performance of the protocol.

Referee #2 (Remarks to the Author):

Review for S.L.N. Hermans et al., "Qubit teleportation between non-neighboring nodes in a quantum network".

In the presented paper, the authors demonstrate a series of quantum information teleportation experiments between non-neighbouring nodes. The manuscript presents a refreshing combination of fundamental physics concepts and technical advances. This allows readers across different fields of research to get a unique insight into the state-of-the-art of quantum networks and the associated challenges.

More specifically, thanks to a series of (well-described) innovative methodological improvements, the authors even show unconditional teleportation with unit efficiency and a fidelity exceeding the classical boundary - although only by two standard deviations.

All in all, the obtained results and methods have to be clearly regarded as a major breakthrough, setting a new state-of-the-art in the field. I thus believe that the manuscript deserves publication in Nature, provided that some minor improvements and clarifications are made, especially regarding point #12 in my comments/suggestions.

The outstanding features of this work are:

* The first-time realisation of a three-node quantum network based on high-level control on five qubits. This result by itself must be regarded as a major breakthrough towards scalable quantum networks.

* The first-time demonstration of unconditional qubit teleportation across non-neighbouring networks nodes. This is a conceptual breakthrough, as it simulates the operation regime in actual multi-node quantum networks in which quantum communication is required beyond simple two-node cases.

* A combination of excellent methodological improvements and innovations, that mainly draft the improvement of local and two-node quantum state fidelities. Although some improvements have been proposed in the Supplementary Information of the authors' previous work, Science 372, 259-264 (2021) (phonon side band detection to reject double-emission events), it is the first time that I see these improvements being implemented in an actual experiment. In addition, their new basis-alternating readout sequence leads to a surprisingly rapid fidelity increase.

The beauty of these techniques is that they are not specific to their system (NV centre in diamond), but that they can be easily implemented to boost the performance across a multitude of other systems. They are thus extremely relevant and valuable.

My further general comments are:

- * The paper is nicely written with a well-structured story line.
- * Abstract, introduction and summary are scientifically sound and appropriate. Introduction could be simplified somewhat (see comment #1).
- * Statements are clearly supported by appropriate references.
- * Methodology and analysis of the experimental results are appropriate.
- * Data and Supporting Information are transparent, allowing experts to verify the quality of the results.
- * I would be happy if the Supplementary Material could be extended with some additional experimental details and justification of some experimental parameters (see below).
- * The experimental complexity and advances presented in the current paper go quite significantly beyond the authors' recent paper, *Science* 372, 259-264 (2021), which is maybe a bit underscored in the manuscript. It is however very refreshing to read a factual paper without over-exaggerations.

In the following, I address my critics and suggestions to the current manuscript:

- 1.) In the introductory part, I would feel happy if the authors could shorten the sentences a bit in order to make it easier for non-experts to follow the presentation of the teleportation protocol.
- 2.) On page 2, top right (line ~148), the authors describe temporal photon filtering with 15 ns (and in Fig. 2d additionally with 7.5 ns). I understand that such windowing can help to reduce (to some extent) the influence of dephasing and laser drifts. However, I miss the information at which point in time the window is opened, e.g., with respect to the point of highest intensity in the excitation pulse? Do the authors observe an improvement in fidelities when the window opening is delayed compared to the excitation pulse? What is the suppression ratio of the (cross-polarised?) laser rejection scheme and how many laser breakthrough photons are detected per excitation attempt?
- 3.) On page 3, in the third paragraph (line 233), the authors state that inconsistent consecutive outcomes are "filtered out". Does this mean that the experiment is aborted in this case, right? If it is so, I would recommend to mention it here. It would also be helpful to specify by how much the overall experimental rate is improved compared to the previous conditioning on only the outcomes in the 0-state.
- 4.) On page 3, in the third paragraph (line 240), the authors claim that infidelities stem from qubit flips during the first readout. Please justify this statement, e.g., with data/simulation.
- 5.) In general, error bars are not defined in the figure captions.

- 6.) In Fig. 3a, the labels n and q are not described, similar in Fig. 4a with the labels m and n .
- 7.) In Fig. 3d, the y-axis label has a typo (Sngle ==> Single), same for Fig. S6a.
- 8.) In the Supporting Information, chapter 3, first paragraph: the authors state a real-time rejection of false events. What is the actual experimental delay?
- 9.) In the Supporting Information, chapter 3, last paragraph: the authors give the values α_i for Alice, Bob and Charlie. Can you motivate the choice of these particular parameters? This reasoning would be helpful for other groups that want to join the field.
- 10.) In the Supplementary Information, there are a couple of full-stops missing, especially after referencing to a Supporting Figure.
- 11.) In the Supplementary Information, Chapters 6 and 8: Is the DOI link now published?
- 12.) In the Supplementary Information, Chapter 8: What was the decision-making process to stop data acquisition after 2272 events? Was the experiment stopped manually by the authors at this point or was this measurement time decided before the experiment was started? I am asking this question because if the decision was made during the experiment with the live-data available to the experimentalists, then there could be the risk of bias. For example, if I play the role of the devil's advocate, then I have to imagine that the experiment was stopped at the point when the data showed the largest deviation from the classical bound (due to statistical fluctuations). Considering that the classical bound was "only" surpassed by 2 standard deviations, I wonder whether this would then weaken results a bit?
I think it is very important that the authors clarify this point to strengthen the impact of the results.
- 13.) In the Supplementary Information, Tables S4 and S5: Can you give an interpretation of the power n of the decays? Why is it different for the different methods?
- 14.) In the Supplementary Information, Figure S7. It is in general very interesting to see the scaling between fidelities and experimental rates, especially for network optimisation. In this sense, I would be very happy if the authors could add a graph below/above that shows the fidelity as a function of the detection window size.
- 15.) In the Supplementary Information, Table S8: The authors assume photon interference visibilities of 0.9 for their simulations. How is this parameter justified? Was it measured and if yes, how? If the associated measurements data far back in time, can it safely be assumed that it remained the same or has it been improved compared to previous work? If yes, how?
- 16.) In the Supplementary Information, Table S8/Conclusions of the main text: Infidelities seem to stem mainly from the mandatory use of non-zero values for α , and the optical phase uncertainty. While α can, in principle, be adjusted sufficiently close to zero, the phase uncertainty seems to remain one of the biggest challenges. Although the authors have developed a very sophisticated scheme in their recent work, Science 372, 259-264 (2021), it seems to me that more improvements

are needed, especially when quantum networks will use fibres that are deployed in harsh and noisy environments. It would be helpful to briefly mention this big challenge (and potential solutions) in the conclusion section. This might trigger more research groups to have a look into this particular problem, such that improved solutions may become available much faster.

Referee #3 (Remarks to the Author):

The authors here present an impressive demonstration of quantum teleportation between non-neighboring NV-centres in a three-node quantum network.

The three nodes, Alice – Bob – Charlie, are operated in the following way. First, communication qubits of Alice and Bob are entangled, then Bob's part of the entanglement is swapped into his memory qubit; second, Bob and Charlie's communication qubits are entangled; Third, Bell-State measurement on Bob's qubits project Alice and Charlie's communication qubits into an entangled state, and Charlie's part of the entanglement is swapped into her memory qubit; Finally, the quantum state of Charlie's qubit is teleported onto Alice's by a Bell-state measurement on Charlie's qubits plus feed-forward.

The authors report an average teleported state fidelity of 70.2%, with an uncertainty of 3 standard deviations above the 2/3 threshold, at a 1/117s rate. Key innovations were implemented to this experiment to enable these impressive results: 1) Rejecting double emission by monitoring the phonon-side band emission improving the fidelity of remote entanglement generation; 2) Memory qubit lifetime enhanced via decoupling pi-pulse allowing beyond 5000 entanglements attempts vs. 1000 without the pi pulses; 3) Basis-alternating repetitive readout of the memory qubits resulting in higher readout fidelity.

Finally, even in the unconditional teleportation regime, the average teleported fidelity is above the 2/3 bound, and with shorter detection window lengths on the two-node entanglement generation, the authors report a 68.8% fidelity at a 1/100s rate (albeit at a reduced clearance above the 2/3 threshold of two standard deviations).

The authors present high-quality data in a clear way with a valid methodology. It is in my opinion that this is done in a detailed way, with many descriptions in the main text, and in particular, the methods and supplemental information are very well detailed. I think the references cover well the quantum networks community. It is in my opinion that this manuscript is suitable for publication in Nature.

I have a few queries:

- (1) The Alice-Bob entanglement fidelity is lower than the Bob-Charlie one, but then after the PSB rejection, it changes. Is this expected given the performance of the relevant NV centers and is there an intuitive explanation here?
- (2) I am curious about what the cause of the model only qualitatively capturing the experimental results in Fig 4b is.
- (3) What are the key areas that need to be addressed to enable more nodes? Are the improvements

demonstrated here enough?

Author Rebuttals to Initial Comments:

Referee #1 (Remarks to the Author):

The report describes a protocol for entangling remote nodes of a quantum network by way of an entanglement swapping procedure at a third (center node). The distributed entanglement is then used for a teleportation operation between the remote nodes. The enabling features of this protocol are the novel quantum engineering schemes used to preserve the fidelity of the distributed entanglement at a critical level for teleportation to be performed above the classical bound. These novel schemes include techniques for removing contributions to the infidelity (false heralding events) of the generated state, preserving adjacent node entanglement long enough for both local links to be fully prepared using efficient and controllable quantum memories at one of the local links. Another key enabling feature is the non-demolition read-out of the quantum memory, thereby preserving the state stored in quantum memory over several readout attempts, thus allowing more attempts to complete a successful entangled link. Subsequently, the authors provide a higher fidelity teleportation ($F = 0.702$ - lower bounded by characterized errors) for a non-deterministic implementation of the protocol and at a slightly lower fidelity ($F=0.688$) and lower rate for a deterministic implementation whereby each communication photon inserted into the network is teleported.

The results are well presented, and the key features/schemes used are well characterized. This helps greatly to follow the progression of the protocol and to see how incremental improvements in performances of various steps in the protocol due to these techniques enable the distributed entanglement to be suitable for teleportation. Importantly, the impact of these innovations (eg applying a decoupling pi-pulse to actively protect the coherence of the memory; feed-forward in the teleportation protocol) are validated by following the particular steps with and without them.

We thank the Referee for their time and their positive comments on our manuscript.

I just have a few questions or clarifications. The technique at the end of the day is similar to a quantum repeater protocol - yet this term is not used. Is there a reason for this or do the authors disagree that the process represents a quantum repeater protocol? Why not refer to this as a quantum repeater protocol since it creates multiple elementary links (Alice to Bob and Bob to Charlie) and then employs entanglement swapping by way of a Bell State Measurement (BSM) at Bob to extend the entanglement to the further away nodes at Alice and Charlie?

We thank the referee for this suggestion to link the entanglement swapping to a quantum repeater protocol. We have revised line 57 to read:

“To prepare the teleporter we use an entanglement swapping protocol mediated by Bob, similar to a quantum repeater protocol [1], to establish entanglement between Alice and Charlie.”

[1] Briegel, H., Dür, W., Cirac, J. I. & Zoller, P. Quantum Repeaters: The Role of Imperfect Local Operations in Quantum Communication. *Phys. Rev. Lett.* **81**, 5932–5935 (1998).

Can you detail what percentage of false heralding events are rejected in the scheme using phonon-side band detection?

The probability to reject a false heralding event equals the probability to detect the additional photon by the phonon-side band (PSB) detector (see main text, lines 132-136), given by the product of the probability that the photon is emitted in the PSB (~ 0.97) and the average detection efficiency

of the PSB detection path (0.10 for link AB and 0.12 for link BC) - yielding the “approx. 10%” mentioned in line 136.

Are there situations where the second photon (the signature of the falsity) can go undetected (other than the obvious failure of the detection event itself). In other words, can the second photon ‘hide’ (perhaps by exiting the BS on the same path as the first photon) so as to be undetected and thus this scheme does not flag them? If so – what impact can this have on the overall performance of the protocol.

The probability that the second photon arrives (‘hides’) at the ZPL detectors is roughly equal to the heralding photon detection probability (see Table S8), which ranges from $2.4e-4$ to $5e-4$. For the current experiment we can thus safely ignore any contribution caused by this possibility.

Referee #2 (Remarks to the Author):

Review for S.L.N. Hermans et al., "Qubit teleportation between non-neighboring nodes in a quantum network".

In the presented paper, the authors demonstrate a series of quantum information teleportation experiments between non-neighbouring nodes. The manuscript presents a refreshing combination of fundamental physics concepts and technical advances. This allows readers across different fields of research to get a unique insight into the state-of-the-art of quantum networks and the associated challenges.

More specifically, thanks to a series of (well-described) innovative methodological improvements, the authors even show unconditional teleportation with unit efficiency and a fidelity exceeding the classical boundary - although only by two standard deviations.

All in all, the obtained results and methods have to be clearly regarded as a major breakthrough, setting a new state-of-the-art in the field. I thus believe that the manuscript deserves publication in Nature, provided that some minor improvements and clarifications are made, especially regarding point #12 in my comments/suggestions.

The outstanding features of this work are:

- The first-time realisation of a three-node quantum network based on high-level control on five qubits. This result by itself must be regarded as a major breakthrough towards scalable quantum networks.
- The first-time demonstration of unconditional qubit teleportation across non-neighbouring networks nodes. This is a conceptual breakthrough, as it simulates the operation regime in actual multi-node quantum networks in which quantum communication is required beyond simple two-node cases.
- A combination of excellent methodological improvements and innovations, that mainly draft the improvement of local and two-node quantum state fidelities. Although some improvements have been proposed in the Supplementary Information of the authors' previous work, Science 372, 259-264 (2021) (phonon side band detection to reject double-emission events), it is the first time that I see these improvements being implemented in an actual experiment. In addition, their new basis-alternating readout sequence leads to a surprisingly rapid fidelity increase.

The beauty of these techniques is that they are not specific to their system (NV centre in diamond), but that they can be easily implemented to boost the performance across a multitude of other systems. They are thus extremely relevant and valuable.

My further general comments are:

- The paper is nicely written with a well-structured story line.
- Abstract, introduction and summary are scientifically sound and appropriate. Introduction could be simplified somewhat (see comment #1).
- Statements are clearly supported by appropriate references.
- Methodology and analysis of the experimental results are appropriate.
- Data and Supporting Information are transparent, allowing experts to verify the quality of the results.

- I would be happy if the Supplementary Material could be extended with some additional experimental details and justification of some experimental parameters (see below).
- The experimental complexity and advances presented in the current paper go quite significantly beyond the authors' recent paper, *Science* 372, 259-264 (2021), which is maybe a bit underscored in the manuscript. It is however very refreshing to read a factual paper without over-exaggerations.

We thank the Referee for their thorough review and their compliments on our results.

In the following, I address my critics and suggestions to the current manuscript:

1.) In the introductory part, I would feel happy if the authors could shorten the sentences a bit in order to make it easier for non-experts to follow the presentation of the teleportation protocol.

We thank the Referee for their suggestion. We have broken up the (very!) long sentence in the introduction (lines 22-28) to yield:

“Using a teleporter in the form of a pre-shared entangled state, the quantum information is transferred by performing a joint Bell-state measurement on the sender’s part of the entangled state and the qubit state to be teleported. The state is recovered on the receiving node by a gate operation conditioned on the Bell-state measurement outcome.”

2.) On page 2, top right (line ~148), the authors describe temporal photon filtering with 15 ns (and in Fig. 2d additionally with 7.5 ns). I understand that such windowing can help to reduce (to some extent) the influence of dephasing and laser drifts. However, I miss the information at which point in time the window is opened, e.g., with respect to the point of highest intensity in the excitation pulse? Do the authors observe an improvement in fidelities when the window opening is delayed compared to the excitation pulse? What is the suppression ratio of the (cross-polarised?) laser rejection scheme and how many laser breakthrough photons are detected per excitation attempt?

To describe the experimental method of temporal photon filtering in more detail, we add a subsection to Section 2 of the Supplementary Materials:

“2.1 Temporal selection of heralding photons

To eliminate any reflected excitation light in the heralding detectors, we make use of a cross-polarization scheme and perform temporal selection of the detected photons as described in Reference [1]. We start the detection windows 4 ns (5 ns) after the highest intensity point of the excitation pulse, for the AB (BC) entangled link, to ensure sufficient suppression of excitation laser light in the detection window.

[1] Bernien, H. *et al.* Heralded entanglement between solid-state qubits separated by three metres. *Nature* **497**, 86–90 (2013).”

3.) On page 3, in the third paragraph (line 233), the authors state that inconsistent consecutive outcomes are “filtered out”. Does this mean that the experiment is aborted in this case, right? If it is so, I would recommend to mention it here. It would also be helpful to specify by how much the overall experimental rate is improved compared to the previous conditioning on only the outcomes in the 0-state.

We would like to answer to this comment in a two-folded manner. First, to clarify the exact methodology of the data acquisition we have extended the section ‘Data acquisition and experimental rates’ in the Supplementary materials:

“At multiple points during the experimental sequence we make a decision on whether to continue the protocol or not. For example, after successful heralding of a two-node entangled state, we can decide to abort the protocol based on whether the flag was raised by any detection of a PSB photon. As all these signals come in real-time, these decisions can be made in real-time, and the sequence can be aborted whenever appropriate. However, we choose to postpone these decisions to the processing after the data acquisition and continue the sequence in any case. In this way, we gain more insight in the performance of the experiment, at the expense of just a marginal increase in experimental time.

Processing steps taken after the data acquisition:

- More precise temporal selection than 15 ns.
- PSB rejection.
- Selection of readout outcomes during the Bell-state measurements, including selection on consistent readout patterns for the memory qubit readout.
- Selection on successful Charge-Resonance checks during the sequence.
- Ensure that the last optical phase measurement (before feedback) prior to the heralding event is below $<50^\circ$.
- High enough photon count rates on Alice during qubit initialization and optical pulsing, averaged over the second before the heralding signal comes in. (On Alice we perform gate tuning to keep the qubit on resonance with the control lasers. The gate tuning, in combination with the high repetition rate of entanglement attempts, makes the qubit spectrally jumpy. The control loops during the charge-resonance checks should ensure the resonance condition [1], but we use this live-tracking of the photon statistics as an extra check.)

To emphasize, all these processing steps can also be done real-time during the experiment. For the data acquisition, we interleave blocks of measurements with calibrations. The calibrations also serve as an independent measure of the performance of the setups.”

[1] Hensen, B. *et al.* Loophole-free Bell inequality violation using electron spins separated by 1.3 kilometres. *Nature* **526**, 682–686 (2015).

Regarding the second point, we would continue the sequence roughly twice as often by accepting two outcomes (“00” and “01”) per Bell-state measurement. Since we do two Bell-state measurements in the sequence, the improvement would be roughly a factor of 4. Using the exact numbers for the readout fidelities and the memory gate fidelities, we find a total improvement of 3.6. This factor is not exactly the improvement in experimental rate, since the second readout of the memory qubit also adds time to the sequence (although small compared to the time it takes to herald both entangled states).

4.) On page 3, in the third paragraph (line 240), the authors claim that infidelities stem from qubit flips during the first readout. Please justify this statement, e.g., with data/simulation.

The data shown in Figures 3d of the main text and S6a in the Supplementary Materials indicate that the memory qubit operations are imperfect and can alter the memory qubit state with small probability. If such an error happens in the first readout repetition in combination with a faulty readout result, this readout error cannot be detected with the second repetition as the initial state is lost. This intuition is validated in the simulations.

To clarify, we revise the sentence on lines 238-240 of the main text: “At this point, the remaining observed infidelity mainly results from cases where the memory qubit was flipped during the first readout block **due to imperfect memory qubit gates.**”

5.) In general, error bars are not defined in the figure captions.

To clarify the definition of the error bars used in the figures we add the following sentence to all data figure captions:

“All error bars represent one standard deviation.”

6.) In Fig. 3a, the labels n and q are not described, similar in Fig. 4a with the labels m and n .

To make the used symbols in the figure more clear, we make the following revisions to the relevant parts of the captions:

Figure 3:

“Upon success in the n^{th} attempt, a phase feed-forward is applied to maintain the correct reference frame of the memory qubit [17], followed by a decoupling pulse on the memory qubit. The decoupling π_m pulse causes a Z-rotation on the communication qubit. Afterwards, we rephase the memory qubit for the same amount of time as it took to herald entanglement (by applying q blocks of XY8 decoupling sequences on the communication qubit, where q depends on the number of entanglement attempts needed n) and we end with another phase feed-forward on the memory qubit, to compensate for any phase picked up during this decoupling.”

Figure 4:

“Circuit diagram of the teleportation protocol using notation defined in Figure 3. $m(n)$ is the number of attempts needed to herald entanglement for the AB (BC) entangled link.”

7.) In Fig. 3d, the y-axis label has a typo (Sngle ==> Single), same for Fig. S6a.

We thank the Referee for their sharp eye. We have corrected the axis labels.

8.) In the Supporting Information, chapter 3, first paragraph: the authors state a real-time rejection of false events. What is the actual experimental delay?

The detection of a false event is flagged by photon detection using the PSB detectors and is therefore similar to heralding entanglement using the ZPL detectors. Hence there is no additional experimental delay related to the real-time rejection. For more details see our reply to comment #3.

9.) In the Supporting Information, chapter 3, last paragraph: the authors give the values α_i for Alice, Bob and Charlie. Can you motivate the choice of these particular parameters? This reasoning would be helpful for other groups that want to join the field.

The values for α used in the two-node entanglement generations are chosen such that we have an optimal trade-off between entanglement generation rate and fidelity. The reason behind the differing settings per experimental setup are due to the differences in detection probabilities, see Table S8. By choosing $\alpha_A p_{\text{det},A} \sim \alpha_B p_{\text{det},B}$, we balance the probabilities for the heralding photon to originate from either setup. We would like to refer to the Supplementary Materials of our recent paper, *Science* 372, 259-264 (2021), for more details.

We add an explicit reference to this paper for clarity.

10.) In the Supplementary Information, there are a couple of full-stops missing, especially after referencing to a Supporting Figure.

We thank the Referee for their eye to detail. We did not manage to find missing full-stops, but we trust that the editor will correct any remaining ones.

11.) In the Supplementary Information, Chapters 6 and 8: Is the DOI link now published?

The DOI link will be published upon publication. All materials that will be posted on the link were attached to the submission.

12.) In the Supplementary Information, Chapter 8: What was the decision-making process to stop data acquisition after 2272 events? Was the experiment stopped manually by the authors at this point or was this measurement time decided before the experiment was started? I am asking this question because if the decision was made during the experiment with the live-data available to the experimentalists, then there could be the risk of bias. For example, if I play the role of the devil's advocate, then I have to imagine that the experiment was stopped at the point when the data showed the largest deviation from the classical bound (due to statistical fluctuations). Considering that the classical bound was "only" surpassed by 2 standard deviations, I wonder whether this would then weaken results a bit?

I think it is very important that the authors clarify this point to strengthen the impact of the results.

We thank the Referee for this question. Following standard policy in our lab, we have not analyzed the data until we finished the data acquisition, indeed to prevent stopping bias. To further clarify our measurement procedure, we have extended section 8 on Data acquisition and experimental rates in the Supplementary materials:

" We collect the data in blocks of 200 "raw" data points (taking roughly an hour), which result on average in about 30 data points per block after applying the processing steps. We analyzed the data only after completing all data acquisition. Prior to the measurement, we decided on the target total number of data points, the experimental settings, and the processing steps afterwards. The plan was to run a sufficient number of blocks (estimated at 80) such that after processing we would remain with >2,800 data points, and at least 450 per cardinal axis. These target numbers were a trade-off between measurement time and expected violation of the classical bound for the conditional teleportation. Unfortunately, after ~80% of the data points were acquired, the setups consistently failed two of the calibrations steps due to the formation of ice on Charlie's diamond sample (the origin of this, either a leak or an outgassing element, is under investigation at time of writing). Therefore, we decided to end the data acquisition, include all data taken up to that point and analyze. In total, we have acquired 79 blocks of data, and we measured 2272 events ($|+X\rangle$ 382, $|-X\rangle$ 385, $|+Y\rangle$ 385, $|-Y\rangle$ 378, $|+Z\rangle$ 375, $|-Z\rangle$ 367) for the conditional teleportation over a time span of 21 days."

13.) In the Supplementary Information, Tables S4 and S5: Can you give an interpretation of the power n of the decays? Why is it different for the different methods?

In general, many factors influence n , including the particular sequence used, and the complex interplay between spin bath-induced decoherence, dephasing due to entanglement attempts, and pulse errors. Determining the exact effects of these factors and providing a full explanation of the power n is beyond the scope of the current work. Instead, we use the fits for an operational description of the system.

What we can say though, as we write in the main text on lines 184-188, is that the shape of the decay of the memory coherence with entanglement attempts and the decoupling π_M pulse (dark blue circles in Fig 3b) is not described solely by (intrinsic) spin bath dephasing, since we would expect a $n = 2$ in that case.

14.) In the Supplementary Information, Figure S7. It is in general very interesting to see the scaling between fidelities and experimental rates, especially for network optimisation. In this sense, I would be very happy if the authors could add a graph below/above that shows the fidelity as a function of the detection window size.

We would like to note that the corresponding fidelities are plotted in Figure 4d of the Main text; to facilitate the reader, Fig. S7 and 4d use the same color scheme.

15.) In the Supplementary Information, Table S8: The authors assume photon interference visibilities of 0.9 for their simulations. How is this parameter justified? Was it measured and if yes, how? If the associated measurements data far back in time, can it safely be assumed that it remained the same or has it been improved compared to previous work? If yes, how?

The estimate of the photon interference visibility is indeed based on previously measured data in our group, see e.g. Humphreys et al., Nature 558, 268-273 (2018). As the relevant experimental parts have not significantly changed, we assume this result is still a good estimate.

16.) In the Supplementary Information, Table S8/Conclusions of the main text: Infidelities seem to stem mainly from the mandatory use of non-zero values for α , and the optical phase uncertainty. While α can, in principle, be adjusted sufficiently close to zero, the phase uncertainty seems to remain one of the biggest challenges. Although the authors have developed a very sophisticated scheme in their recent work, Science 372, 259-264 (2021), it seems to me that more improvements are needed, especially when quantum networks will use fibres that are deployed in harsh and noisy environments. It would be helpful to briefly mention this big challenge (and potential solutions) in the conclusion section. This might trigger more research groups to have a look into this particular problem, such that improved solutions may become available much faster.

We thank the Referee for this suggestion. We have added the following sentence to the outlook on line 352:

“In addition, future work will focus on further improving the phase stabilization and extending the current schemes for use in deployed fiber [1].”

[1] Grein, M. E., Stevens, M. L., Hardy, N. D. & Benjamin Dixon, P. Stabilization of long, deployed

optical fiber links for quantum networks. *2017 Conf. Lasers Electro-Optics, CLEO 2017 - Proc.* **2017-January**, 1–2 (2017).

Referee #3 (Remarks to the Author):

The authors here present an impressive demonstration of quantum teleportation between non-neighboring NV-centres in a three-node quantum network.

The three nodes, Alice – Bob – Charlie, are operated in the following way. First, communication qubits of Alice and Bob are entangled, then Bob's part of the entanglement is swapped into his memory qubit; second, Bob and Charlie's communication qubits are entangled; Third, Bell-State measurement on Bob's qubits project Alice and Charlie's communication qubits into an entangled state, and Charlie's part of the entanglement is swapped into her memory qubit; Finally, the quantum state of Charlie's qubit is teleported onto Alice's by a Bell-state measurement on Charlie's qubits plus feed-forward.

The authors report an average teleported state fidelity of 70.2%, with an uncertainty of 3 standard deviations above the 2/3 threshold, at a 1/117s rate. Key innovations were implemented to this experiment to enable these impressive results: 1) Rejecting double emission by monitoring the phonon-side band emission improving the fidelity of remote entanglement generation; 2) Memory qubit lifetime enhanced via decoupling pi-pulse allowing beyond 5000 entanglements attempts vs. 1000 without the pi pulses; 3) Basis-alternating repetitive readout of the memory qubits resulting in higher readout fidelity.

Finally, even in the unconditional teleportation regime, the average teleported fidelity is above the 2/3 bound, and with shorter detection window lengths on the two-node entanglement generation, the authors report a 68.8% fidelity at a 1/100s rate (albeit at a reduced clearance above the 2/3 threshold of two standard deviations).

The authors present high-quality data in a clear way with a valid methodology. It is in my opinion that this is done in a detailed way, with many descriptions in the main text, and in particular, the methods and supplemental information are very well detailed. I think the references cover well the quantum networks community. It is in my opinion that this manuscript is suitable for publication in Nature.

We would like to thank the Referee for their time and the positive words on our manuscript.

I have a few queries:

(1) The Alice-Bob entanglement fidelity is lower than the Bob-Charlie one, but then after the PSB rejection, it changes. Is this expected given the performance of the relevant NV centers and is there an intuitive explanation here?

The fidelity of the Bob-Charlie entangled link is higher than the Alice-Bob entangled link after PSB rejection (see Figure 2d), due to two reasons. 1) The average alpha for the Bob-Charlie link is higher than the average alpha of the Alice-Bob link, see Table S8. This increases the probability for a double $|0\rangle$ state occupancy error. Hence, we expect a larger improvement by introducing the PSB rejection. 2) The average PSB detection efficiency of the Bob-Charlie entangled link is higher (see Table S8), resulting in a higher probability to detect the additionally emitted photon.

(2) I am curious about what the cause of the model only qualitatively capturing the experimental results in Fig 4b is.

We would like to emphasize that the data and the model are also quantitatively consistent; please note that the measurement uncertainty depicted is just 1 s.d. (we forgot to indicate this definition in the captions in the original manuscript). In particular, the model overlaps with the 2 s.d. interval for all the data in Fig 4b and 4c.

(3) What are the key areas that need to be addressed to enable more nodes? Are the improvements demonstrated here enough?

The main current limitation to the network is the number of remote entangled pairs that can be created per node, which is bounded by decoherence of the memory qubit during remote entangling attempts. A major goal is therefore to reach (as we write in the outlook) “the threshold where nodes can reliably deliver a remote entangled state while preserving previously stored quantum states in their memory qubits”. This will require improvements in the entangling rate (e.g. by an improved optical interface) and/or on the memory qubit (see e.g. our recent results towards improved memories: Robust quantum-network memory based on spin qubits in isotopically engineered diamond, <http://arxiv.org/abs/2111.09772>).

Reviewer Reports on the First Revision:

Referees' comments:

Referee #1 (Remarks to the Author):

The updated manuscript addresses all the minor concerns/issues I had with the original. I now recommend it for publication.

Referee #2 (Remarks to the Author):

The authors answered to all my comments and I am very happy with the improvements made on the manuscript, especially with the more detailed description on data acquisition and analysis (previous comment #12).

Therefore I can now recommend publication of the manuscript as it is.

Reviewer #3 provided confidential comments to the editor, recommending publication.